# Comparative Analysis of Mitogenomes among Five Species of *Filchnerella* (Orthoptera: Acridoidea: Pamphagidae) and Their Phylogenetic and Taxonomic Implications

**DOI:** 10.3390/insects12070605

**Published:** 2021-07-02

**Authors:** Fang-Yuan Zheng, Qiu-Yue Shi, Yao Ling, Jian-Yu Chen, Bo-Fan Zhang, Xin-Jiang Li

**Affiliations:** 1The Key Laboratory of Zoological Systematics and Application, Institute of Life Sciences and Green Development, College of Life Sciences, Hebei University, Baoding 071002, China; zhengfangyuan313@163.com (F.-Y.Z.); Lingyao0112@163.com (Y.L.); chenjianyu0618@163.com (J.-Y.C.); zbf0710@outlook.com (B.-F.Z.); 2National Engineering Laboratory for Lake Pollution Control and Ecological Restoration, State Environmental Protection Key Laboratory for Lake Pollution Control, State Environmental Protection Scientific Observation and Research Station for Lake Dongtinghu, State Environmental Protection Key Laboratory of Estuarine and Coastal Environment, Chinese Research Academy of Environmental Sciences, Beijing 100012, China; qyshi@iue.ac.cn

**Keywords:** Acridoidea, Pamphagidae, *Filchnerella*, mitogenomes, phylogeny, wing length

## Abstract

**Simple Summary:**

*Filchnerella* belongs to Insecta, Orthoptera and Pamphagidae, of which there are 19 recorded species. The wings of *Filchnerella* are diverse, including three grasshopper wing types: longipennate, short wings and small wings. The previous studies of *Filchnerella* are more about the description of species morphology, and few about exploring the phylogenetic relationships with limited number of species and DNA fragments, which are insufficient to study the phylogeny of the entire genus, especially in order to understand the evolution of wing types in *Filchnerella*. To better understand the mitogenomic characteristics of *Filchnerella* and reveal its internal phylogenetic relationships, the complete mitochondrial genomes of *Filchnerella sunanensis*, *Filchnerella amplivertica*, *Filchnerella dingxiensis*, *Filchnerella pamphag**oides* and *Filchnerella nigritibia* were sequenced and comparatively analyzed in this study. The mitogenomes of these five *Filchnerella* species were found to be highly conserved. Phylogenetic analyses, based on mitogenome data of 16 species of Pamphagidae, using both the maximum likelihood (ML) and Bayesian inference (BI) methods, supported the monophyly of *Filchnerella* and produced valuable data for the phylogenetic study of the genus.

**Abstract:**

Mitogenomes have been widely used for exploring phylogenetic analysis and taxonomic diagnosis. In this study, the complete mitogenomes of five species of *Filchnerella* were sequenced, annotated and analyzed. Then, combined with other seven mitogenomes of *Filchnerella* and four of Pamphagidae, the phylogenetic relationships were reconstructed by maximum likelihood (ML) and Bayesian (BI) methods based on PCGs+rRNAs. The sizes of the five complete mitogenomes are *Filchnerella sunanensis* 15,656 bp, *Filchnerella amplivertica* 15,657 bp, *Filchnerella nigritibia* 15,661 bp, *Filchnerella pamphagoides* 15,661 bp and *Filchnerella dingxiensis* 15,666 bp. The nucleotide composition of mitogenomes is biased toward A+T. All tRNAs could be folded into the typical clover-leaf structure, except that tRNA Ser (AGN) lacked a dihydrouridine (DHU) arm. The phylogenetic relationships of *Filchnerella* species based on mitogenome data revealed a general pattern of wing evolution from long wing to increasingly shortened wing.

## 1. Introduction

The genus *Filchnerella* Karny, 1908, belongs to Insecta, Orthoptera, Acridoidea and Pamphagidae, and is the largest genus in the family Pamphagidae, with 19 known species [1], accounting for nearly 1/3 of all the Pamphagidae species in China. The genus is endemic to China and is distributed in the arid northwestern provinces of Gansu, Qinghai, Ningxia and northern Shaanxi. The wing length is an important taxonomic character of *Filchnerella* [1]. The wings length of the genus includes longipennate (Tegmina very long, extending beyond the end of hind femora in male and reaching to or extending beyond the posterior of the third abdominal tergite in female and their length equal to the pronotum), short wings (Tegmina shortened, length distinctly greater than the length of pronotum, but the apex distinctly not reaching the end of hind femora in male, and distinctly shorter than metazona; the apex reaching, not reaching or slightly extending beyond the posterior margin of the second abdominal tergite in female) and small wings (Tegmina strongly abbreviated, length distinctly shorter than the length of pronotum in male and metazona in female). Different wing types are a widespread phenomenon in Orthoptera insects and have evolutionary consequences [2]. According to the perspective of evolutionary taxonomy, wings length is of great significance in the evolution of grasshoppers. The long wing is an ancestral characteristic, the shortened wing is an evolutionary characteristic and the wing-lessness is the newest characteristic, which is based on the overall evolutionary trend of Acridoidea [3]. Chen et al. [4] studied the phylogenetic implications in wing type evolution of Catantopidae. Does wing type evolution of species within one genus follow the above pattern? This lacks explicit reporting on grasshoppers. *Filchnerella* is a preferred material to study the question because of their diverse wing types.

The degree of wing development has a close relationship with insects’ movement ability. Mitochondria, as the main place of aerobic respiration in most eukaryotes [5], support the various life activities of individuals, such as flight. Insect mitochondrial genome is a circular molecule structure with the size of 15 to 18 kb, which contains 13 protein-coding genes (PCGs), 22 transfer RNA genes (tRNAs), two ribosomal RNA genes (rRNAs) and A+T-rich region [6,7,8]. Mitogenome has many unique advantages for some molecular systematic studies: the extraction of the mitogenome is easier than nuclear genes, and the genome is small, with a high copy number; there are few intercalated spacers, and strict matrilineal inheritance avoids the randomness of parental inheritance. In addition, although the sequence of the mitogenome is highly conserved and its structure is stable, the evolution rate of the mitogenome sequence is 5–10 times higher than that of nuclear genome. Therefore, mtDNA has become an important material for the study of phylogeny [9,10].

With the development of PCR technology and DNA sequencing technology, the usage of DNA sequences to study the phylogeny and evolution of insects has become a popular method [4,11,12,13,14,15]. There are also a few reports on the related studies of Pamphagidae. Flook et al. [16,17] studied the 12S rRNA and 16S rRNA sequence of three Pamphagidae species from Africa when studying the molecular phylogeny of the Orthoptera. Li et al. [18] studied the genetic differentiation among different populations of *Haplotropis brunneriana* and three other species of grasshoppers from China. Zhang et al. [19] carried out a molecular systematic analysis of some genera of Pamphagidae from China based on partial sequences of 16S rDNA. Zhang et al. [20] studied molecular phylogeny of Pamphagidae from China based on mitochondrial cytochrome oxidase II sequences. Zhang et al. [21] studied the complete mitochondrial genomes of three Pamphagidae grasshoppers: *Asiotmethis zacharjini*, *Filchnerella helanshanensis* and *Pseudotmethis rubimarginis*.

In *Filchnerella*, most existing studies are description of species based on morphology, while a few explored the phylogenetic relationship of *Filchnerella* using molecular methods. Li et al. [22] studied the phylogenetic relationships of six species of *Filchnerella* based on partial sequence of 16S rDNA. Zhang et al. [23] analyzed the molecular characteristics in the partial sequence of the mitochondrial COII gene of eight *Filchnerella* species. These studies only focused on seldom species and DNA fragments of *Filchnerella*, which is insufficient to study the phylogeny of the entire genus, especially in order to understand the evolution of wing types in the genus.

In this study, the mitogenomes of five species of *Filchnerella* were newly sequenced, annotated and analyzed. The phylogenetic relationship was reconstructed based on mitogenome data of twelve *Filchnerella* species, with four additional species of Pamphagidae as outgroups. It focuses on the evolution trend of wing length in *Filchnerell* and provides molecular data to the systematic study.

## 2. Materials and Methods

### 2.1. Sample and DNA Extraction

The grasshoppers used in the study are showed in Table 1 and Table 2. The samples were originally conserved in 95% ethanol, and then transferred to 4 °C for cryopreservation. The Mitochondrial DNA were extracted from the hind femora muscle. Mitochondrial DNA was separated according to the method of Tamura and Aotsuka [24] with modifications [25,26].

### 2.2. PCR Amplification and Sequencing

The complete mitogenomes sequences of other species of *Filchnerella* previously obtained in our laboratory and the complete mitogenomes sequences of Pamphagidae downloaded from GenBank were used as reference sequences for identifying conservative regions in ClustalX1.83 [31]. Primers used in the present study including the general primer, referring to Zhi et al. [29], and specific primer designed using Primer Premier 5.0(PREMIER Biosoft, San Francisco, CA, USA) [32] and DNAMAN 6.0(Lynnon Biosoft., San Ramon, CA, USA) [33]. PCR reactions [28,29,30] were carried out under the following conditions: 5 min initial denaturation at 94 °C, followed by 30 cycles of 30 s at 94 °C, annealing at 50 °C for 30 s, elongation at 72 °C for 30 s and a final elongation for 5 min at 72 °C. All samples were submitted to Sangon Biotech (Shanghai) Co., Ltd. (Shanghai, China). for sequencing.

### 2.3. Sequence Assembly, Annotations and Analysis

The sequences files obtained by sequencing were opened with Seqman software to proofread for repeats at the beginning and end of the sequence. Then, DNAMAN 6.0 [33] was used to concatenate all the sequences obtained. The tRNAscan-SE 1.21 was used to identify tRNA in the concatenate complete mitochondrial genome sequence, including the relative position, length, anticodon and secondary structure of tRNA. A+T rich regions and rRNA genes were identified by comparing with published mitochondrial genome sequences of grasshoppers from the family Pamphagidae. Relative synonymous codon usage (RSCU) of PCGs was calculated in MEGA X [34]. The ratio of nonsynonymous substitution (Ka) to synonymous substitution (Ks) for all PCGs was calculated using DnaSP 5.0 [35].

### 2.4. Phylogenetic Analysis

Phylogenetic analyses were performed on the dataset of 13 PCGs and two rRNAs from 16 complete mitogenomes of Pamphagidae, including the five mitogenomes of *Filchnerella* that were sequenced in the present study. Four species in three other Pamphagidae genera, *Humphaplotropis culaishanensis*, *Thrinchus schrenkii*, *Asiotmethis jubatus* and *Asiotmethis zacharjini,* were used as outgroups. The combined sequence of protein-coding and rRNA encoding genes was aligned by ClustalX1.83 (Conway Institute UCD Dublin, Dublin, Ireland) [31]. ModelFinder (Research School of Biology, Australian National University, Canberra, Australia) [36] was used to select the best-fit model using AICc criterion. Maximum likelihood (ML) analysis was conducted using IQ-TREE (http://www.iqtree.org/, accessed on 20 March 2021) [37] in PhyloSuite v1.2.2 (Key laboratory of Aquaculture Disease Control, Wuhan, Hubei, China) [38]. Node confidence was assessed with 1000 bootstrap replicates. Bayesian inference (BI) analysis was executed with 10 million generations, sampling trees every 1000 generations in MrBayes 3.1.4 (http://nbisweden.github.io/MrBayes/index.html, accessed on 20 March 2018) [39], and selected GTR model (lset nst = 6 rates = invgamma). The consensus tree was calculated after discarding the first 2500 trees (25%) as the burn-in phase.

## 3. Results and Discussion

### 3.1. Genome Structure

The complete mitogenomes of *Filchnerella sunanensis* Liu, 1982; *Filchnerella amplivertica* Li, Zhang & Yin, 2009; *Filchnerella nigritibia* Zheng, 1992; *Filchnerella pamphag**oides* Karny, 1908; and *Filchnerella dingxiensis* Zhang, Xiao & Zhi, 2011, were sequenced, and their length was 15,656 bp, 15,657 bp, 15,661 bp, 15,661 bp and 15,666 bp, respectively (Table 2). Their structures are the same as those of insects [4,40]. Each mitogenome was found to be composed of circular double-stranded molecules, containing the typical set of 37 genes (13 typical protein-coding genes (PCGs), 22 transfer RNA genes (tRNAs), two ribosomal RNA genes (rRNAs)) and an A+T rich region (Figure 1). Most of these genes were located on the J-strand (9 PCGs and 14 tRNAs), whereas the other genes (4 PCGs, 8 tRNAs and 2 rRNAs) were located on the N-strand (Table 3).

### 3.2. Nucleotide Composition

The nucleotide composition of five grasshopper species revealed a strong A+T bias in the whole mitogenome (Table 4). The A+T content ranged from 72.3% (*F. nigritibia*) to 72.9% (*F. sunanensis*). A+T % was the highest in the A+T-rich region (83.4%) and the lowest in tRNA genes (69.6%), which is consistent with the characteristic of base composition bias in insect mitogenomes [4,22,23,40]. Strand bias was demonstrated in nucleotide compositional skew. The skew statistics indicated that the whole mitogenome, the rRNAs, the tRNAs and the A+T rich region of the five species of *Filchnerella* were negative for the GC-skew and positive for the AT-skew. The PCGs of the five species of *Filchnerella* were negative for the GC-skew and negative for the AT-skew.

### 3.3. Protein-Coding Genes and Codon Usage

The 13 PCGs within five species of *Filchnerella* ranged from 162 bp (*atp8*) to 1717 bp (*nd5*) in size (Table 3). The size and arrangement are conserved. The biased usage of A/T could also be reflected in codon frequencies. The A+T content of all protein codons is higher than that of G+C, and each base of the codon indicated that the A+T content of the last site was higher than that of the first two sites. This is consistent with the characteristics of high A+T bias of base composition and selection pressure from G+C to A+T in insect mitochondrial genomes. All the initiation codons in the mitogenomes of the five species of *Filchnerella* were ATN, and ATG was the most frequently used initiation codon. Most termination codons were TAN, and TAA were the most frequently used termination codon.

Relative synonymous codon usage (RSCU) of five species of *Filchnerella* is shown in Figure 2, and for most amino acids, the usages of synonymous codons are biased. In addition, the synonymous codon preferences are conserved. In the five species of *Filchnerella*, Ala, Arg, Gly, Leu, Pro, Ser, Thr and Val are the most frequently encoded amino acids. Similarly, the biased usage of A+T nucleotides is also reflected by RSCU. The most frequently used codons are TTA, ATT, TTT and ATA, indicating the preference of nucleotide A/T in the five species of *Filchnerella*.

### 3.4. Transfer and Ribosomal RNA Genes

The size of 22 transfer RNAs (tRNA) of five *Filchnerella* grasshoppers ranged from 63 bp to 72 bp (Table 3). Among them, 21tRNAs could be folded into the typical clover-leaf structure, except that tRNA Ser (AGN) lacked a dihydrouridine (DHU) arm (Figure 3). The lengths of the acceptor stem (7 bp) and anticodon loop (7 bp, except for the trnA of *F. amplivertica* (9 bp)) are conserved. The classic secondary structures comprised a DHU arm (2–4 bp), a TψC arm (3–6 bp) and an anticodon arm (3–6 bp). However, the extra arm and loops of DHU and TψC were more variable, with obvious nucleotide substitutions and length variation. Additionally, noncanonical match of G-U and mismatches of U-U, U-C, A-A, A-G and A-C were scattered throughout tRNA stems. Among the five *Filchnerella* grasshoppers, mitochondrial, rrnL-encoding and rrnS-encoding genes were oriented on the N-strand and located at the conserved positions between *trnL1 (CUN)* and *trnV*, and between *trnV* and the A+T-rich region, respectively. The rrnL ranged from 1314 bp (*F. pamphagoides*) to 1321 bp (*F. dingxiensis*) in size, while the rrnS varied from 851 bp (*F. amplivertica*) to 854 bp (*F. pamphagides*). Therefore, there was no substantial size variation between rRNAs within five *Filchnerella* mitogenomes.

### 3.5. Ka/Ks of 12 Grassoppers of Filchnerella

To characterize the evolutionary patterns of 13 PCGs, the Ka/Ks across the 12 *Filchnerella* mitogenomes were calculated. As shown in Figure 4, similar to previous studies in insects [41,42], the Ka/Ks value for *nad4L* is the highest, followed by the *nad6* and *atp8*; the lowest value is for *cox1*. The Ka/Ks values for all PCGs are <1, indicating that they are not neutral and are evolving under purifying selection. The gene *nad4L* (Ka/Ks = 0.32812) exhibits the highest rate, which is suggested to be under the least selection pressure [43] and the fastest evolving gene among the mitochondrial PCGs in *Filchnerella*. The gene *cox1* (Ka/Ks = 0.02182) exhibits the smallest rate, which has been regarded as under strong purifying selection [43,44,45].

### 3.6. Phylogenetic Analysis

The same tree topologies were recovered from both BI and ML analyses with high bootstrap values (BS) and Bayesian posterior probability values (PP) in the most clades (Figure 5). The included *Filchnerella* species were shown to form a monophyletic clade. Although the BS value (50%) is relatively low and the branching structure may be unstable, the PP value (0.99) is very high, which strongly supports this topology structure. Therefore, this branch structure is still credible, which is consistent with the traditional taxonomic result [1].

The tree showed that *Filchnerella* can be divided into three major clades. *F. sunanensis* separated first at the base of the tree, forming an independent clade A. Clade B is the sister to clade C, with high BS (100%) and PP (1). Nine species *Filchnerella* were clustered into clade B. *F. pamphagides* and *F. dingxiensis* were clustered into clade C.

Clade A (*F. sunanensis*) separated first from *Filchnerella*, which is consistent with Li et al. [22]. *F. sunanensis* has the longest tegmina in *Filchnerella* (extends far beyond the end of hind femora in male), which is the most ancestral species in traditional evolutionary taxonomy [3]. It is supported by molecular systematics results in the present study.

In clade B, first, *F. qilianshanensis* (tegmina extending just beyond the end of hind femora in male) and *F. kukunoris* (tegmina extending the base of epiproct) were clustered together at the base of the clade. Then, *F. tenggerensis*, *F. beicki* (tegmina extending the base of epiproct), *F. helanshanensis* (tegmina not reaching the 1/2 of hind femora), *F. amplivertica* (tegmina not reaching the base of epiproct), *F. nigritibia* (tegmina extending the 1/3 of hind femora), *F. yongdengensis* (tegmina extending the 1/2 of hind femora) and *F. rubrimargina* (tegmina not reaching the 1/2 of hind femora) separate from the ancestor of clade B in order. Except for *F. helanshanensis* and *F. yongdengensis*, the evolutionary trend of wing length in clade B follows the direction of wing length evolution in grasshoppers (from long wing to short wing) [3]. Therefore, although only females were found, we also can infer that the length of tegmina of *F. tenggerensis* in males extends the base of the epiproct.

Clade C comprises 2 species: *F. pamphag**oides* and *F. dingxiensis*, and both are small wings whose tegmina is strongly abbreviated, and the length is distinctly shorter than the length of the pronotum in males. According to the traditional taxonomic view, clade C should separate from clade B, but it is a parallel evolution with clade B in the present study. This may be related to the unique origin of the small wings. The phylogenetic relationship of the 12 species of *Filchnerella* grasshoppers that resulted from this study shows similarities compared with those based on morphology and molecular data revealed in previous studies [22,23,46].

## 4. Conclusions

In this study, the complete mitochondrial genome of *F. sunanensis*, *F. amplivertica*, *F. nigritibia*, *F. pamphagides* and *F. dingxiensis* were sequenced, annotated and analyzed. It was found that their structures are the same as those of Acridoidea. The nucleotide composition of five grasshopper species revealed a strong A+T bias in the complete mitogenome. Relative synonymous codon usage (RSCU) of five species of *Filchnerella* shows that the usages of synonymous codons are biased for most amino acids. All tRNAs could be folded into the typical clover-leaf structure, except that tRNA Ser (AGN) lacked a dihydrouridine (DHU) arm and there was no substantial size variation between rRNAs within five *Filchnerella* mitogenomes. The Ka/Ks values for all PCGs are <1, indicating that they are evolving under purifying selection. These five newly sequenced mitogenomes of genus *Filchnerella* can provide valuable data for future studies of phylogenetic relationships of Pamphagidae. The phylogenetic tree showed that the evolutionary relationships within *Filchnerella* were monophyletic clade, and the evolutionary trend of *Filchnerella* well demonstrated the direction of wing length evolution.

## Figures and Tables

**Figure 1 insects-12-00605-f001:**
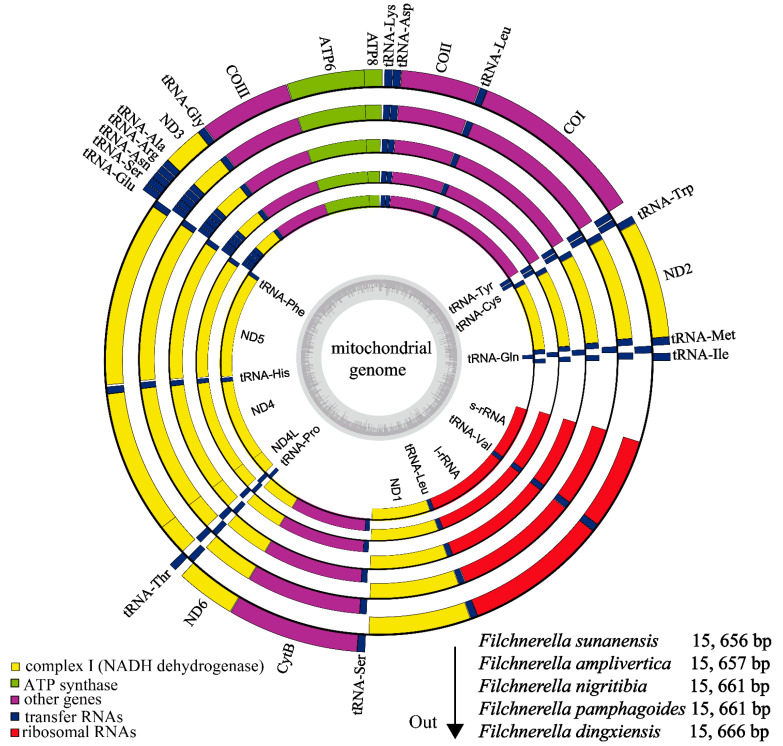
The circular maps of five species of grasshopper mitogenomes. Genes are characterized by different color blocks. The J-strand is visualized on the outer circle and the N-strand on the inner circle. The species from inside to outside are as follows: *F. sunanensis*, *F. amplivertica*, *F. nigritibia*, *F. pamphagoides* and *F. dingxiensis*.

**Figure 2 insects-12-00605-f002:**
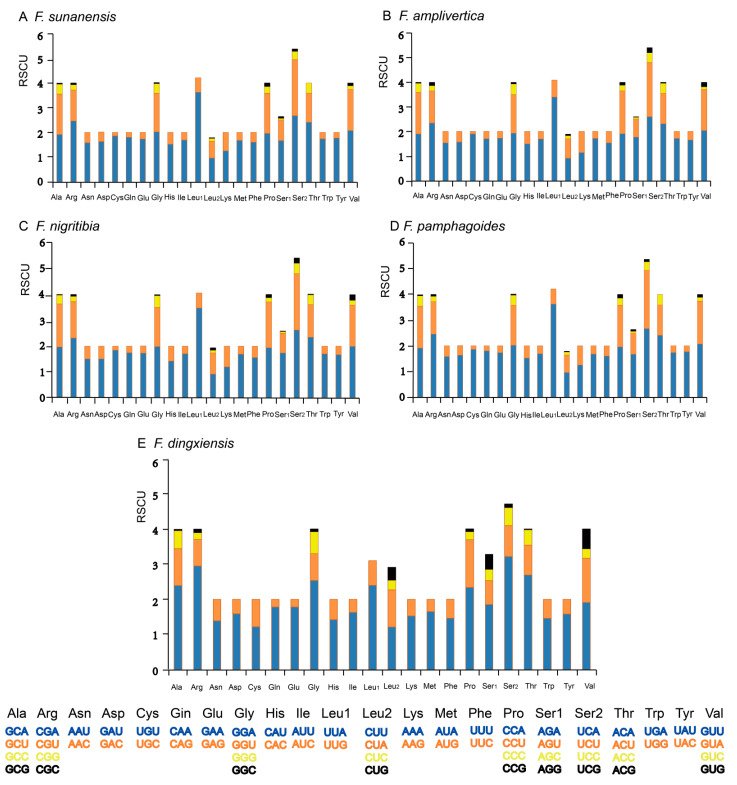
Relative synonymous codon usage (RSCU) of five grasshoppers of *Filchnerella*.

**Figure 3 insects-12-00605-f003:**
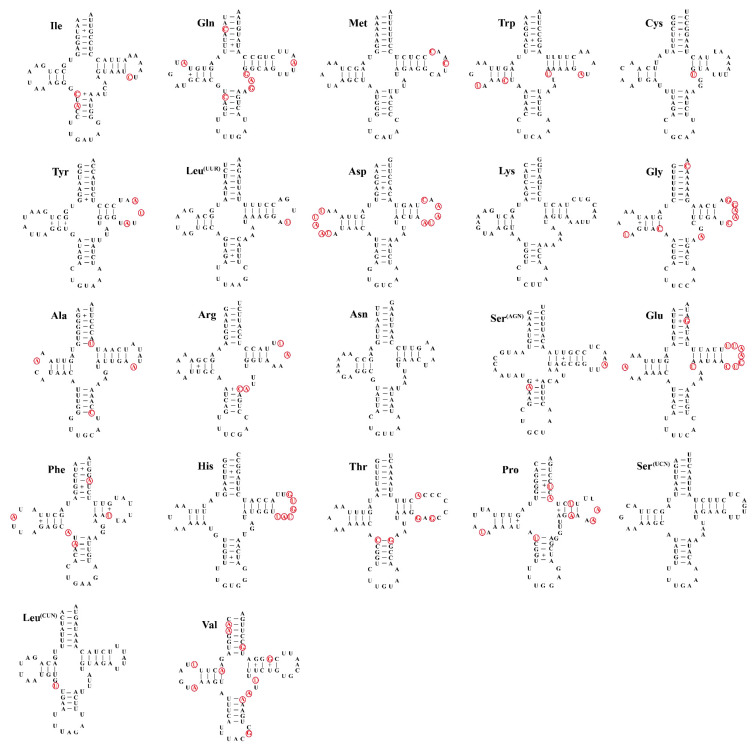
Secondary structures of 22 tRNAs identified in the mitogenome of *F. sunanensis*. All of the genes are shown in order of occurrence. Watson–Crick base pairings and mismatches are represented by dashes (-) and pluses (+). The conserved and variable sites among the other 4 species of grasshoppers are indicated with black and red hollow circles, respectively.

**Figure 4 insects-12-00605-f004:**
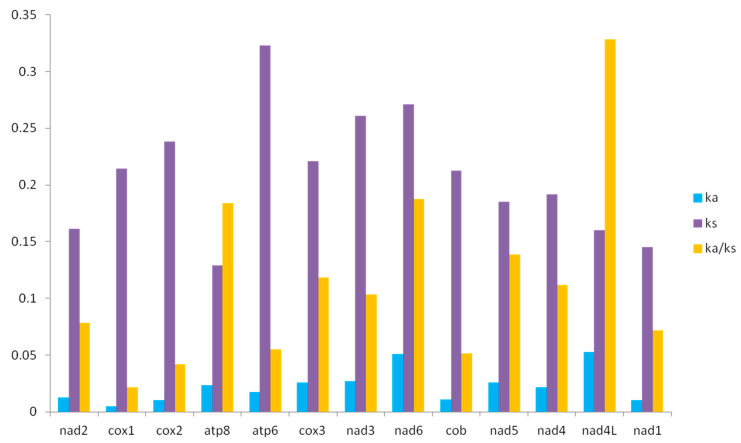
Synonymous (Ka) and non-synonymous (Ks) substitutional rates and the ratios of Ka/Ks of PCGs in 13 grasshoppers of *Filchnerella*.

**Figure 5 insects-12-00605-f005:**
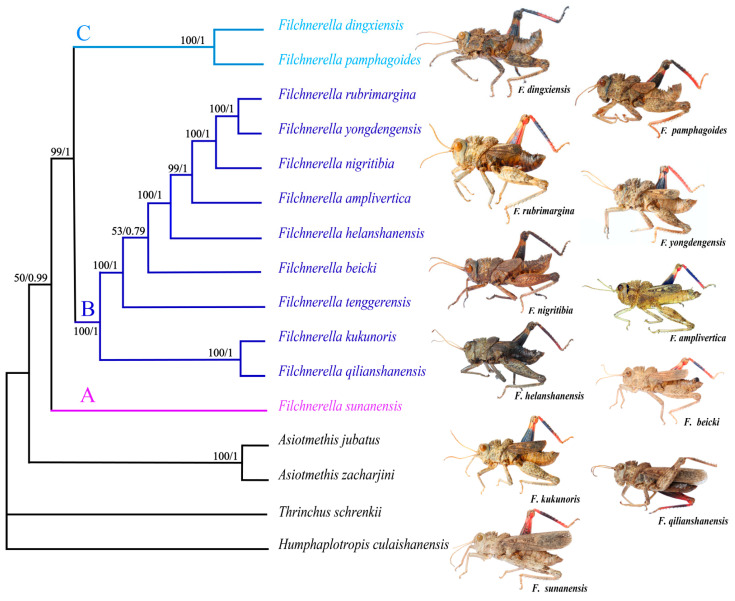
Phylogenetic relationships of 16 grasshoppers based on the concatenated nucleotide sequences of PCGs + rRNAs. Numbers on branches were Bootstrap values (BS, **left**) and Bayesian posterior probabilities (PP, **right**).

**Table 1 insects-12-00605-t001:** Information on collecting samples.

Species	Collecting Time	Collecting Sites	Collector
*Filchnerella amplivertica* Li, Zhang & Yin, 2009	July 2003	Zhongwei, Ningxia	LI Xin-Jiang, WANG Wen-Qiang
*Filchnerella pamphag**o**ides* Karny, 1908	September 2003	Lanzhou, Gansu	ZHANG Dao-Chuan, LI Xin-Jiang
*Filchnerella sunanensis* Liu, 1982	July 2006	Sunan, Gansu	LI Xin-Jiang, ZHENG Jin-Yu
*Filchnerella nigritibia* Zheng, 1992	July 2006	Dawukou, Ningxia	ZHANG Dao-Chuan, LI Xin-Jiang
*Filchnerella dingxiensis* Zhang, Xiao & Zhi, 2011	July 2006	Dingxi, Gansu	ZHANG Dao-Chuan, ZHI Yong-Chao

**Table 2 insects-12-00605-t002:** Summary of mitogenomes used in this study.

Species	Size (bp)	Accession No	Reference
*Filchnerella nigritibia* Zheng, 1992	15,661	MZ433420	This study
*Filchnerella amplivertica* Li, Zhang & Yin, 2009	15,657	MZ433418	This study
*Filchnerella pamphag**oides* Karny, 1908	15,661	MZ433417	This study
*Filchnerella sunanensis* Liu, 1982	15,656	MZ433421	This study
*Filchnerella dingxiensis* Zhang, Xiao & Zhi, 2011	15,666	MZ433419	This study
*Filchnerella beicki* Ramme, 1931	15,658	NC_024923	[27]
*Filchnerella helanshanensis* Zheng, 1992	15,657	NC_020329	[21]
*Filchnerella qilianshanensis* Xi & Zheng, 1984	15,661	NC_046558	Unpublished
*Filchnerella tenggerensis* Zheng & Fu, 1989	15,659	NC_046559	Unpublished
*Filchnerella**rubrimargina* Zheng, 1992	15,661	NC_052733	Unpublished
*Filchnerella**yongdengensis* Xi& Zheng, 1984	15674	MK_903560	[2]
*Filchnerella**kukunoris* Bey-Bienko, 1948	15662	MK_903590	[2]
*Humphaplotropis culaishanensis* Li, Cao & Yin, 2014	15,659	NC_023535	[28]
*Thrinchus schrenkii* Fischer von Waldheim, 1846	15,672	NC_014610	[29]
*Asiotmethis jubatus* (Uvarov, 1926)	15,669	NC_025904	[30]
*Asiotmethis zacharjini* (Bey-Bienko, 1926)	15,660	NC_020328	[21]

**Table 3 insects-12-00605-t003:** Annotations for the five species grasshopper mitogenomes.

Gene	Coding Strand	Nucleotide Number	Size (bp)	Intergenic Length	Anticodon	Initiation Codon	Termination Codon
*tRNA^Ile^*	J	1–68/./././.	68/./././.	0/./././.	GAT/./././.		
*tRNA^Gln^*	N	69–137/./././.	69/./././.	−1/./././.	TTG/./././.		
*tRNA^Met^*	J	137–205/./././.	69/./././.	0/./././.	CAT/./././.		
*ND2*	J	206–1228/./././.	1023/./././.	2/./././.		ATG/./././.	TAG/./././.
*tRNA^Trp^*	J	1231–1297/././1231–1298/.	67/././68/.	−8/./././.	TCA/./././.		
*tRNA^Cys^*	N	1290–1354/././1291–1355/.	65/./././.	8/11/9/8/.	GCA/./././.		
*tRNA^Tyr^*	N	1363–1428/1364–1429/././.	66/./././.	1/./././.	GTA/./././.		
*COI*	J	1430–2963/1431–2964/././.	1534/./././.	0/./././.		CCG/./././.	T/./././.
*tRNA^Leu(UUR)^*	J	2964–3029/2965–3030/././.	66/./././.	4/././5/4	TAA/./././.		
*COII*	J	3034–3717/3035–3718 /3035–3718/3036–3719/.	684/./././.	3/./././.		ATG/./././.	TAA/./././T
*tRNA^Asp^*	J	3721–3789/3722–3791/./3723–3793/.	69/70/./71/.	2/./././.	GTC/./././.		
*tRNA^Lys^*	J	3792–3862/3794–3864/3794–3864 /3796–3866/3796–3866	71/./././.	17/./././.	CTT/./././.		
*ATP8*	J	3880–4041/3882–4043/./3884–4045/.	162/./././.	−7/./././.		ATC/./././.	TAA/./././.
*ATP6*	J	4035–4712/4037–4714/./4039–4716/.	678/./././.	3/./././4		ATG/./././.	TAA/./././.
*COIII*	J	4716–5507/4718–5509/. /4720–5511/4721–5512	792/./././.	3/./././.		ATG/./././.	TAA/./././.
*tRNA^Gly^*	J	5511–5577/5513–5580/./5515–5581/5516–5582	67/68/./67/.	0/./././.	TCC/./././.		
*ND3*	J	5578–5931/5581–5932/5581–5934 /5582–5935/5583–5936	354/352/ 354/./.	−2/0/−2/./.		ATC/./././.	TAG/T /TAG/./.
*tRNA^Ala^*	J	5930–5995/5933–5998/. /5934–5999/5935–6000	66/./././.	0/3/././.	TGC/./././.		
*tRNA^Arg^*	J	5996–6059/6002–6064/6002–6065 /6003–6066/6004–6067	64/63/64/./.	2/1/./2/.	TCG/./././.		
*tRNA^Asn^*	J	6062–6126/6066–6130/6067–6131 /6069–6133/6070–6134	65/./././.	0/./././.	GTT/./././.		
*tRNA^Ser (AGN)^*	J	6127–6193/6131–6197/6132–6198 /6134–6200/6135–6201	67/./././.	1/./././.	GCT/./././.		
*tRNA^Glu^*	J	6195–6263/6199–6266/6200–6267 /6202–6268/6203–6269	69/68/./67/.	1/./././.	TTC/./././.		
*tRNA^Phe^*	N	6265–6329/6268–6333/6269–6334 /6270–6335/6271–6336	65/66/././.	0/./././.	GAA/./././.		
*ND5*	N	6330–8046/6334–8050/6335–8051 /6336–8052/6337–8053	1717/./././.	15/./././.		ATT/./././.	T/./././.
*tRNA^His^*	N	8062–8131/8066–8132/8067–8133 /8068–8135/8069–8136	70/67/./68/.	0/./−1/./.	GTG/./././.		
*ND4*	N	8132–9465/8133–9466/8133–9467 /8135–9469/8136–9470	1334/./1335/./.	−7/./././.		ATG/./././.	TA/./TAG/./.
*ND4L*	N	9459–9752/9460–9753/9461–9754 /9463–9756/9464–9757	294/./././.	2/./././.		ATG/./././.	TAA/./././.
*tRNA^Thr^*	J	9755–9822/9756–9823/9757–9826 /9759–9826/9760–9827	68/./70/68/.	0/./././.	TGT/./././.		
*tRNA^Pro^*	N	9823–9888/9824–9889/9827–9893 /./9828–9894	66/./67/./.	2/./././.	TGG/./././.		
*ND6*	J	9891–10412/9892–10413/9896–10417 /9896–10417/9897–10418	522/./././.	3/./././.		GTG/./././.	TAA/./././.
*Cytb*	J	10416–11558/10417–11557/10421–11563 /./10422–11564	1143/1141/ 1143/./.	−2/0/−2/./.		ATG/./././.	TAG/T /TAG/./.
*tRNA^Ser (UCN)^*	J	11557–11626/11558–11627/11562–11631/ /./11563–11632	70/./././.	26/./././25	TGA/./././.		
*ND1*	N	11653–12597/11654–12598/11658–12602 /./.	945/./././.	3/./././4		ATA/./././.	TAG/./././.
*tRNA^Leu(CUN)^*	N	12601–12666/12602–12667/12606–12671 /./12607–12672	66/./././.	0/././3/0	TAG/./././.		
*lrRNA*	N	12667–13986/12668–13987/12672–13991 /12675–13988/12673–13993	1320/./. /1314/1321	0/././2/0			
*tRNA^Val^*	N	13987–14056/13988–14058/13992–14062 /13991–14061/13994–14064	70/71/././.	0/./././.	TAC/./././.		
*srRNA*	N	14057–14908/14059–14909/14063–14914 /14062–14914/14065–14917	852/851/852/853/.	0/./././.			
A+T rich		14909–15656/14910–15657/14915–15661 /./14918–15666	748/./747/./.				

**Note:** N and J indicate that the gene was located in the minor (N) and major (J) strand. “/.”: same as the one previous to it. The species are as follows: *F. sunanensis, F. amplivertica, F. nigritibia, F. Pamphag**oides* and *F. dingxiensis*.

**Table 4 insects-12-00605-t004:** Nucleotide composition features of the 5 species of *Filchnerella*.

Species	Regions	T%	C%	A%	G%	Size (bp)	A+T%	AT-Skew	GC-Skew
*F. sunanensis*	All genes	30.6	16.1	42.3	11.0	15,656.0	72.9	0.16	−0.19
	rRNAgenes	29.7	16.9	43.2	10.3	2172.0	72.9	0.19	−0.24
	tRNA genes	31.1	16.5	38.7	13.6	1482.0	69.8	0.11	−0.10
	A+T-rich region	37.7	9.4	45.6	7.4	748.0	83.3	0.09	−0.12
	PCGs	41.4	14.0	31.2	13.5	11,182.0	72.6	−0.14	−0.02
	All codons								
	1st	35	13.4	31.7	20.3	3729.0	66.7	−0.05	0.20
	2nd	45	20.4	19.7	14.6	3727.0	64.7	−0.39	−0.17
	3rd	44	8.1	42.1	5.6	3726.0	86.1	−0.02	−0.18
	Genes on J-strand								
	1st	28	15.8	35.1	20.6	2298.0	63.1	0.11	0.13
	2nd	43	22.6	20.5	13.5	2297.0	63.5	−0.35	−0.25
	3rd	32	12.0	53.2	2.4	2297.0	85.2	0.25	−0.67
	Total	34.8	16.8	36.3	12.2	6892.0	71.1	0.02	−0.16
	Genes on N-strand								
	1st	45	9.5	26.1	19.7	1431	71.1	−0.27	0.35
	2nd	48	16.9	18.5	16.4	1430	66.5	−0.44	−0.02
	3rd	63	1.8	24.3	10.6	1429	87.3	−0.44	0.71
	Total	52.1	9.4	23.0	15.6	4290	75.1	−0.39	0.25
*F. amplivertica*	All genes	30.3	16.4	42.1	11.2	15,657.0	72.4	0.16	−0.19
	rRNA genes	29.7	16.9	43.2	10.3	2171.0	72.9	0.19	−0.24
	tRNA genes	31.4	16.5	38.4	13.8	1482.0	69.8	0.10	−0.09
	A+T-rich region	36.9	9.9	46.5	6.7	748.0	83.4	0.12	−0.19
	PCGs	41.1	14.3	30.7	13.9	11,178.0	71.8	−0.14	−0.01
	All codons								
	1st	34	13.6	31.2	20.8	3729.0	65.2	−0.04	0.21
	2nd	45	20.5	19.6	14.7	3725.0	64.6	−0.39	−0.16
	3rd	44	8.8	41.4	6.2	3724.0	85.4	−0.03	−0.17
	Genes on J-strand								
	1st	28	16.2	34.6	21.1	2298.0	62.6	0.11	0.13
	2nd	43	22.7	20.3	13.8	2295.0	63.3	−0.36	−0.24
	3rd	31	13.4	52.1	3.4	2295.0	83.1	0.25	−0.60
	Total	34.2	17.4	35.7	12.8	6888.0	69.9	0.02	−0.15
	Genes on N-strand								
	1st	44	9.6	25.9	20.4	1431	69.9	−0.26	0.36
	2nd	48	17.1	18.5	16.3	1430	66.5	−0.44	−0.02
	3rd	64	1.3	24.1	10.8	1429	88.1	−0.45	0.79
	Total	52.0	9.3	22.8	15.8	4290	74.8	−0.39	0.26
*F. nigritibia*	All genes	30.3	16.5	42.0	11.3	15,661.0	72.3	0.16	−0.19
	rRNA genes	29.7	16.8	43.1	10.4	2172.0	72.8	0.18	−0.24
	tRNA genes	31.7	16.2	38.4	13.7	1486.0	70.1	0.10	−0.08
	A+T-rich region	37.8	10.4	44.4	7.4	747.0	82.2	0.08	−0.17
	PCGs	40.9	14.4	30.9	13.9	11,183.0	71.8	−0.14	−0.02
	All codons								
	1st	34	13.7	31.3	20.7	3729.0	65.3	−0.04	0.21
	2nd	45	20.6	19.7	14.7	3727.0	64.7	−0.39	−0.17
	3rd	43	8.7	41.7	6.2	3727.0	84.7	−0.02	−0.17
	Genes on J-strand								
	1st	28	16.4	34.5	21.1	2298.0	62.5	0.10	0.13
	2nd	43	22.9	20.3	13.8	2297.0	63.3	−0.36	−0.25
	3rd	31	13.3	52.4	3.3	2297.0	83.4	0.26	−0.60
	Total	34.0	17.5	35.8	12.7	6892.0	69.8	0.03	−0.16
	Genes on N-strand								
	1st	44	9.4	26.0	20.1	1431	70.0	−0.26	0.36
	2nd	48	17.1	18.7	16.2	1430	66.7	−0.44	−0.03
	3rd	63	1.4	24.4	10.8	1430	87.4	−0.44	0.77
	Total	52.0	9.3	23.0	15.7	4291	75.0	−0.39	0.26
*F. pamphag* *oides*	All genes	30.5	16.3	42.0	11.1	15,661.0	72.5	0.16	−0.19
	rRNA genes	30.0	17.2	42.1	10.7	2167.0	72.1	0.17	−0.23
	tRNA genes	30.9	16.4	38.9	13.7	1485.0	69.8	0.11	−0.09
	A+T-rich region	36.1	10.2	46.2	7.5	747.0	82.3	0.12	−0.15
	PCGs	41.3	14.1	30.9	13.7	11,183.0	72.2	−0.14	−0.01
	All codons								
	1st	35	13.3	31.2	20.9	3729.0	66.2	−0.06	0.22
	2nd	45	20.7	19.5	14.8	3727.0	64.5	−0.40	−0.17
	3rd	44	8.3	42.0	5.4	3727.0	86.0	−0.02	−0.21
	Genes on J-strand								
	1st	28	15.9	34.5	21.2	2298.0	62.5	0.10	0.14
	2nd	43	22.9	20.3	13.7	2297.0	63.3	−0.36	−0.25
	3rd	32	12.9	52.2	2.8	2297.0	84.2	0.24	−0.64
	Total	34.5	17.2	35.7	12.6	6892.0	70.2	0.02	−0.15
	Genes on N-strand								
	1st	45	9.2	25.8	20.3	1431.0	70.8	−0.27	0.38
	2nd	48	17.2	18.3	16.4	1430.0	66.3	−0.45	−0.02
	3rd	64	1.0	25.5	9.7	1430.0	89.5	−0.43	0.81
	Total	52.2	9.2	23.2	15.5	4291.0	75.4	−0.38	0.26
*F. dingxiensis*	All genes	30.6	16.3	42.0	11.0	15,666.0	72.7	0.16	−0.19
	rRNA genes	29.9	17.2	42.5	10.3	2175.0	72.4	0.17	−0.25
	tRNA genes	30.9	16.6	38.7	13.8	1484.0	69.6	0.11	−0.09
	A+T-rich region	37.1	9.7	45.7	7.5	750.0	82.8	0.10	−0.13
	PCGs	41.3	14.0	31.0	13.6	11,184.0	72.4	−0.14	−0.01
	All codons								
	1st	35.8	16.3	28.8	19.0	3731.0	64.7	−0.11	0.08
	2nd	50.9	14.4	22.1	12.6	3727.0	73.0	−0.39	−0.07
	3rd	37.3	11.3	42.2	9.3	3726.0	79.4	0.06	−0.10
	Genes on J-strand								
	1st	28.4	15.7	35.4	20.5	2300.0	63.7	0.11	0.13
	2nd	43.3	22.6	20.5	13.7	2297.0	63.8	−0.36	−0.25
	3rd	32.8	12.5	52.3	2.5	2296.0	85.1	0.23	−0.67
	Total	34.8	16.9	36.0	12.2	6893.0	70.9	0.02	−0.16
	Genes on N-strand								
	1st	44.4	9.4	26.0	20.1	1431.0	70.4	−0.26	0.36
	2nd	47.8	17.2	18.4	16.6	1430.0	66.2	−0.44	−0.02
	3rd	63.0	1.4	24.8	10.8	1430.0	87.8	−0.44	0.77
	Total	51.7	9.3	23.1	15.9	4291.0	74.8	−0.38	0.26

**Note:** rRNA genes: ribosomal RNA genes; tRNA genes: transfer RNA genes; PCGs: protein-coding genes; 1st, 2nd, 3rd: the 1st, 2nd, 3rd codon position of the PCGs.

## Data Availability

The data supporting the findings of this study are openly available in National Center for Biotechnology Information (https://www.ncbi.nlm.nih.gov, accessed on 1 July 2021), accession numbers were MZ433420, MZ433418, MZ433417, MZ433421, MZ433419.

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
