# Peer review of "Comparative Analysis of Mitogenomes among Five Species of Filchnerella (Orthoptera: Acridoidea: Pamphagidae) and Their Phylogenetic and Taxonomic Implications"

_insects, 2021, doi:10.3390/insects12070605_

Round 1

Reviewer 1 Report

It is a well-written paper. I would like to suggest it to be accepted. But before that, I think the discussion part can be expanded. The result of the phylogenetic analyses can be applied in several directions. The morphological characters could be discussed as well.

Reviewer 2 Report

Congratulations on a well done project analyzing the mitogenomes of five grasshopper species! The technical part of the genome analyses were well executed, for which I only had minor editorial comments (see attached file). My major concerns are over the interpretation of wing evolution over the resulted phylogenetic relationship of the genus Filchnerella.

  1. The categorization of the wings based on length is problematic without referring to functions, especially regarding short and small wings. It seems to be more logical to categorize the wings into three categories: a. longipennate, b. short, but functional, c. short and nonfunctional. According to this classification, F. Helangshanensis and F. beicki will share the same wing type as F. dingxiensis and F. pamphagides, rather than with F. amplivertica and F. tengerensis.
  2. In the evolutionary process, the loss of flight function has obviously occurred independently trice. Although a trend of wing evolution from full wing to short wing to nonfunctional wing can be seen in the clade comprising F. qilianshanensis through F. nigritibia in the phylogenetic tree (Figure 5), nonfunctional wing has also evolved close to the base directly from full wing in the (F. dingxiensis and F. pamphagides) clade. It is possible that intermediate forms may exist, but not present among the sampled species for this clade, but even then it would not conflict with the pattern revealed in the present analysis, i.e., nonfunctional wing is a homoplasy and can evolve independently, and relatively easily.

In addition, the introduction should expanded to include more recent references on wing size variation in Orthoptera, especially the well-known wing polymorphism in various Gryllus species - for a traits highly variable in a closely related group, it seems likely for flight loss to have occurred multiple times in a lineage. The authors could thus formulate the hypothesis to be tested using the mitogenomic data. 

Reviewer 3 Report

In this manuscript, Zheng et al. applied mitogenomic data to study the phylogenetic relationships among some species of the Orthopteran genus Filchnerella. In addition, the five newly sequenced genomes are described for their molecular features and included in the final tree.

Apart from some minor remarks (highlighted along the text of the returned pdf file) I have some general comments about the manuscript:

1) The manuscript should be corrected for the language by a mother tongue in English. Some parts are not fluent and difficult to read;

2) The phylogenetic analyses are not adequately described and do not conform with standards. In this respect, no mention of the method applied to select the appropriate evolutionary model of nucleotide substitution was provides. I suggest using Partition finder to establish the best model and to partition the data set according to strands, codon positions and marker type. Then BI and ML analyses could be run again;

3) In the result section (and also in part of the introduction) there is only one discussed morphological character, that is wing shape and length. I’m not an expert of grasshopper’s taxonomy, but I guess at species level there must be several other morphological characters to be considered for the general comparison. Is the evolution of wings the only valuable character that deserve to be considered for a phylogenetic context? In addition, In order to complete

4) There is not yet a GenBank accession number for the new mtDNAs, so that if the GenBank Staff could hypothetically change some annotation of the mtDNA’s genes, some reported data (i.e. sequence length and nucleotide frequencies) may be not correct. In the data base there are five deposited complete mitogenomes for the genus: Filchnerella rubrimargina, Filchnerella qilianshanensis, Filchnerella tenggerensis, Filchnerella beicki and Filchnerella helanshanensis. Why is the first species not included in the analyses? I suggest the authors to include this taxon as well for better knowledge of internal relationships of the genus. Generally speaking, there are 11 Pamphagidae mitogenomes available in GenBank. It seems to me a good idea to include all of them phylogenetic completeness.

Reviewer 4 Report

The study of Zheng et al. represents a good attempt to a classical molecular study. They aim to describe five mitochondrial genomes of Filchnerella species (Orthoptera), and test their phylogenetic position.

The study is well designed, but I have specific comments:

l. 48: “and is” instead of “which is”

l. 60: “degree of evolution” is a term that can be misinterpreted

l. 95: what kind of “molecular systematics theories” did you used?

l. 115: can you share the details of the primers you used?

l. 137: you need to justify the model you choose and how many partitions have been taken into account in your analyses?

Round 2

Reviewer 3 Report

In my second review, I noted that the quality of the manuscript has significantly improved. There are still some minor corrections, that I have listed along the text with track changes. After these corrections have been done, I agree with the approval for publication
